# Uremia-Associated Ageing of the Thymus and Adaptive Immune Responses

**DOI:** 10.3390/toxins12040224

**Published:** 2020-04-03

**Authors:** Michiel GH Betjes

**Affiliations:** Department of Internal Medicine, Division of Nephrology and Transplantation, Erasmus Medical Centre, Rg5, P.O. Box 2040, 3000 CA Rotterdam, The Netherlands; m.g.h.betjes@erasmusmc.nl

**Keywords:** uremia, chronic kidney disease, thymus, adaptive immunity, lymphopenia, immunological ageing

## Abstract

Progressive loss of renal function is associated with a series of changes of the adaptive immune system which collectively constitute premature immunological ageing. This phenomenon contributes significantly to the mortality and morbidity of end-stage renal disease (ESRD) patients. In this review, the effect of ESRD on the T cell part of the adaptive immune system is highlighted. Naïve T cell lymphopenia, in combination with the expansion of highly differentiated memory T cells, are the hallmarks of immunological ageing. The decreased production of newly formed T cells by the thymus is critically involved. This affects both the CD4 and CD8 T cell compartment and may contribute to the expansion of memory T cells. The expanding populations of memory T cells have a pro-inflammatory phenotype, add to low-grade inflammation already present in ESRD patients and destabilize atherosclerotic plaques. The effect of loss of renal function on the thymus is not reversed after restoring renal function by kidney transplantation and constitutes a long-term mortality risk factor. Promising results from animal experiments have shown that rejuvenation of the thymus is a possibility, although not yet applicable in humans.

## 1. Introduction

End-stage renal disease (ESRD), with or without renal replacement therapy, is associated with a decreased function of the adaptive immune system, which is known as uremia-associated immune deficiency. This acquired immune deficiency contributes to the increased morbidity and mortality of patients with ESRD [1].

The immune system is classically divided into an adaptive and innate part. Phagocytes, like neutrophils and macrophages, are key players of the innate immune system and have a non-specific modus operandi by phagocytosis and or secreting cytotoxic molecules in response to invading micro-organisms.

Lymphocytes, such as T and B cells, typically belong to the adaptive immune system as they respond to specific elements of the bacteria or virus and build up an antigen-specific memory. This allows for a rapid response of highly dedicated immune cells in case of an antigenic re-challenge (e.g., repeated exposure to the antigen by reinfection). Collaboration between both the adaptive and innate immune system yields maximum effectivity of the adaptive immune response and leads to expansion of antigen-specific T and B cells and the formation of memory cells. Antigen-inexperienced cells, so-called naïve T cells, are constantly generated by the thymus and each T cell has a specific receptor (T cell receptor) for a unique antigen on the cell surface [2]. Antigen-presenting cells, like macrophages and dendritic cells (belonging to the innate immune system), are needed to show the antigen to the T cell receptor (TCR), by presenting it with their HLA molecules and providing co-stimulation for effective activation of the T cell [3]. In addition, the tissue cells themselves may interact with the immune system by the expression of molecular pattern recognition receptors. This class of receptors responds to damage or invading pathogens and allows for the activation of a variety of both immune and non-immune cells. This is relevant to the uremia-associated pro-inflammatory response, as the products of increased oxidative stress, such as advanced glycation end-products (AGE) or oxidized low-density lipoprotein, may activate cells via this family of receptors [4]. 

Loss of renal function is classified in stages of chronic kidney disease (CKD), based on the level of glomerular filtration rate (GFR). Stage 5 of CKD (GFR < 15 mL/min) is synonymous with ESRD and affects almost all parts of both the innate and adaptive immune system. The T cell system in ESRD patients, mostly on dialysis treatment and with little remnant renal function, has been studied most frequently and all studies performed consistently show changes that are compatible with premature immunological ageing [1,5,6,7,8]. Loss of thymic function as well as expansion of memory T cells are central features, both of which are likely involved in susceptibility for infection, decreased vaccination rate, the increase risk for malignancies and atherosclerotic vascular disease [9,10,11].

The possible central role of loss of the thymus function is emerging and discussed.

## 2. The Immune System and Ageing

### 2.1. The Adaptive Immune System

Ageing is associated with a progressive decrease in the functional capacity of the immune system, as reflected in a weakened vaccination response, a higher susceptibility for infections and a decreased tumor surveillance, contributing to an increased incidence of malignancies [12,13,14,15,16,17,18]. In addition, the dysregulation of the immune system increases the risk for autoimmune diseases. 

In parallel with this progressive loss in functional capacity, there may be an increase in the pro-inflammatory status, as evidenced by slightly elevated serum C-reactive protein levels, increased concentrations of serum IL-6 and TNF-alfa, and an expansion of pro-inflammatory T cell population [19]. This combination of events has been coined inflammaging and is considered to contribute to the morbidity and mortality in elderly patients [20].

Although it is likely that ageing may affect all parts of the immune system, most research has focused on the adaptive immune response as this can be studied in great detail by existing techniques and relevant ageing-related changes can be easily detected.

Cells belonging to the adaptive immune system, like B cells and T cells, have a specific cell surface receptor, which is unique for a given T or B cell clone. New T cells with specific TCRs are generated in the thymus by a process of positive and negative selection, ensuring that T cells can respond to antigen-presenting cells but deleted if they are autoreactive and could cause autoimmune diseases. This process is not perfect and as an extra layer of control, regulatory T cells harnessing autoreactive T cells are generated in the thymus as well [21,22]. Thymus epithelial cells (TEC) are instrumental in educating the naïve T cells. The T cells leaving the thymus into the circulation are called naïve T cells or antigen-inexperienced T cells. They can be recognized as recent thymic emigrants (RTE) as they carry a circular DNA in their cytosol, which is a remnant of TCR formation. This is called a TCR excision circle (TREC). The TRECs do not replicate during cell division and are gradually lost in the daughter cells after several rounds of cell proliferation. Measuring TREC content by PCR yields an estimate of recent thymic emigrants in the circulation and indirectly of the thymus function. The expression of the CD31 molecule on the cell surface of RTE by flowcytometry is an easy alternative to TREC measurement and correlates well with this assay [7,23,24,25]. After antigen-specific stimulation via the TCR, the T cell switches to a memory T cell phenotype. With progressive differentiation to an effector-memory T cell, specific effector functions such as rapid cytokine production and cytotoxicity are acquired. Memory T cells will undergo a variable number of repetitive rounds of T cell proliferation depending on the strength of the antigenic stimulation and by repeated exposure to e.g., a virus. This will lead to a decrease in telomere length of their DNA strands and cells may become progressively differentiated and even senescent [26]. Senescent T cells are characterized by a lack of proliferative response to an appropriate antigenic stimulus and expression of cell surface markers facilitating apoptosis. The decrease in telomere length of circulating T cells is, similar to the decline in RTE, almost linearly associated with increase in age, although there is substantial inter-individual variation [7]. 

Optimal antigenic stimulation of lymphocytes requires at least signaling via the antigen receptor in combination with co-stimulatory molecules, but is further enhanced by so-called “danger” signals like DNA fragments and parts of the bacterial cell wall (e.g., lipopolysaccharide) [27]. Dendritic cells are the professional antigen-presenting cells showing the immunogenic peptide within the context of the HLA-molecule on their cell surface to the TCR of the T cells and providing the necessary co-stimulatory signals. Although they are themselves not adaptive immune cells, they are pivotal for an adequate adaptive immune response. 

B cells have the membrane bound immunoglobuline as their B cell receptor. Similar to the TCR, the diversity of the immunoglobulines is substantial. After antigenic stimulation, the naïve B cell develops into a memory B cell and further differentiates to a plasma cell able to secrete large amounts of specific immunoglobulines. Plasma cells migrated to the bone marrow constitute a long-lived population of cells providing lasting humoral immunity. Optimal differentiation of B cells to plasma cells is supported by interactions with helper T cells (CD4 positive T cells) and dendritic cells [28].

### 2.2. Ageing-Related Decrease in Thymus Function

The hallmark of an ageing adaptive immune system is the gradual involution of the thymus with increasing age (reviewed in [29]). This leads to a decreased output of naïve T cells, which can be monitored by the numbers of circulating CD31 positive naïve T cells or alternatively the TREC content in the T cell population. The decrease in thymic output has an almost linear relationship with age despite large inter-individual variation and is used in forensic medicine for age determination [30]. The naïve T cell population is largely maintained by the increased homeostatic proliferation of naïve T cells, specifically within the CD4 T cell population [31,32]. The number of CD8 naïve T cells in circulation does not seem to be maintained by homeostatic proliferation to the same extent as CD4 naïve T cells and becomes very low in the elderly [33]. The proportion of regulatory T cells being produced by the thymus may remain stable or show a relative increase [29]. The decreased thymic output of newly formed naïve T cells will lead to a decrease in the width of the TCR repertoire and diminishes the potential response to pathogens [34]. 

Two hypotheses have been proposed for the ageing-related loss of thymic output. The “soil” hypothesis states that stromal niches for the development of precursor T cells, both in the bone marrow and the thymus, decrease with age [35,36]. The “seed” hypothesis suggests that there is a shift toward myeloid precursor cells instead of lymphoid precursor cells, thereby leaving the innate immunity (granulocytes, monocytes, etc.,) intact but decreasing the number of newly formed lymphocytes [37,38]. For both hypotheses, experimental evidence exists, and they are not mutually exclusive. Experiments infusing precursor T cells from an aged animal into a young animal showed that old precursor cells may function normally in a young environment, indicating that the possible influence of ageing on the precursor cells is essentially reversible once an unaffected stromal niche is offered [34,39,40].

### 2.3. Ageing-Related Expansion of Memory T Cells

In contrast to the naïve T cell compartment, the memory T cell compartment may expand during ageing and contain more and more highly differentiated memory T cells like the Temra cells [41]. These memory T cells have regained the CD45RA isoform of the common leucocyte marker CD45, which is expressed on naïve T cells. The more differentiated memory T cells are expressing more pro-inflammatory cytokines and progressively lose important co-stimulatory cell surface molecules like CD27 and CD28 [42,43]. The telomere length of these cells is usually considerably shortened and oligoclonality in the T cell repertoire may ensue [44]. The inflation of the memory T cell pool may also contribute to an impaired immune response by occupying the “immunological space” (the total number of cell niches available in e.g., bone marrow) available [45]. The lifetime exposure to pathogens is driving the expansion of memory T cells. In this respect, cytomegalovirus infection stands out as it may leave a sizeable footprint in the pool of circulating memory T cells of 10% or more [46,47]. In addition, after CMV latency has been established there is a slight but measurable increase in the pro-inflammatory status [48]. The inflated pool of memory T cells is thought to contribute to the pro-inflammatory status of elderly individuals and may contribute to ageing-related health problems such as Alzheimer disease and cardiovascular disease [49,50,51].

## 3. Renal Failure and Adaptive Immunity

Lymphopenia is a typical finding in individuals with ESRD [52,53] and the extent of lymphopenia is strongly associated with the stage of chronic kidney disease [26]. The increase in virus-associated cancers, tuberculosis and decreased vaccination response to T cell dependent antigens in ESRD patients suggest a major clinical role for impairment of T cells [11].

Similar to individuals with normal renal function an important underlying mechanism is the decreased production naïve T cells because of thymus involution [7,26,52]. In addition, the naïve T cells of ESRD patients have an increased expression of interleukin-2 receptors (CD25) and express a pro-apoptotic profile, which makes them prone for activation-induced cell death [7]. This mechanism of peripheral deletion can contribute to a decreased number of total naïve T cells as it will lower the efficacy of the homeostatic proliferative response, which compensates for thymus involution. The current assays for identifying RTE cannot distinguish between increased turnover and deletion of naïve T cells or decreased thymic output as both processes will lead to a decreased number of CD31positive naïve T cells or TREC content. ESRD patients with a very low number of naïve T cells showed virtually no compensatory increase in CD31 negative naïve T cells which may indicate that activation-induced cell death is a major contributing factor in these patients [54]. 

It is not known which mechanisms contribute to the uremia-associated decrease in thymic function. The uremic milieu could affect the hematopoietic stem cells and promote a shift to the myeloid cell lineage, resulting in less seeding of the thymus by lymphocyte progenitor cells by local inflammation and/or oxidative stress leading to less lymphoid progenitor cells [38]. However, fatty degeneration of the thymus by early senescence of thymus stromal cells could also be a mechanism as a result of inflammation and oxidative stress and malnutrition [55]. 

The memory T cell pool is variably changed in ESRD patients. The absolute number of CD4 T cells is on average decreased because of the decline in naïve T cells and central-memory T cells [55]. Particularly under the influence of CMV latency, the otherwise rare population of cells having lost the pivotal CD28 co-stimulatory molecule may be greatly expanded [43,47,51,56]. The finding of an expanded CD4CD28 negative T cell population is a hallmark of CMV latency irrespective of the presence of CKD [57,58]. But this population is on average significantly larger in CKD patients and may comprise over 50% of all circulating CD4 T cells [47]. Remarkably not all CMV seropositive individuals have such an expansion of CD4CD28 negative T cells, which is likely related to the initial infectious dose and the number of CMV reactivations [51,59,60,61,62]. The total number of CD8 T cells is less affected by ageing, as the decline in naïve CD8 T cells is compensated by an expansion of highly differentiated memory T cells. Strikingly, after CMV infection the circulating memory CD8 T cell population is inflated twofold, which can be largely attributed to expansion of highly differentiated effector memory and Temra cells [47,60]. The premature ageing of T cells in CKD patients is dependent on the calendar age, degree of GFR loss and CMV serostatus, which are all intricately interwoven [63,64,65,66]. 

In both CD4 and CD8 populations, there is an ageing-related loss of telomere length because of the proliferative history of the cells. The T cell telomere length declines linearly with age and is on average significantly lower in ESRD patients than in healthy individuals [7,8].

With different techniques, it was shown that the TCR repertoire of patients with ESRD was skewed because of oligoclonal expansion of specific T cell populations [67,68]. In addition, the PERK phosphorylation pathway in CD4 T cells was disturbed, analogous to what has been described in elderly healthy individuals [69]. Finally, not only intrinsic T cell factors may be involved but also environmental factors, as T cell proliferation after polyclonal stimulation is poor if performed in uremic serum [7] and the function of regulatory T cells also have been shown to be diminished [1]. The T cell composition in the secondary lymphoid organs of ESRD patients has been studied and in accordance with studies in individuals with normal renal function showed little presence of effector T cells [25]. There was a high correlation between numbers of naïve T cells in the lymph nodes and within the peripheral blood, indicating that the ESRD-related lymphopenia of naïve T cells is probably not caused by redistribution of these cells.

Similar to T cells, a decrease in naïve B cells in the total circulating B cell population is found in ESRD patients and these B cells are also prone for apoptosis [70]. However, the B cell and the differentiation from B cell into an immunoglobulin secreting plasma cell has not been thoroughly studied in ESRD patients. 

ESRD also impact the total numbers and function of DC [9]. The density of DC in the skin may be decreased, thereby reducing the capacity to present antigens in the skin (e.g., vaccination antigens) to T cells in the draining lymph node. In addition, the subset of plasmoid DC is decreased in relation to the loss of GFR, while the myeloid DC remain relatively unaffected [9,71,72,73,74]. Stimulating DC function with GM-CSF resulted in improved serological responses of hemodialysis patients to a hepatitis B surface antigen vaccine [9].

In summary, progressive loss of renal function affects naïve T cell generation, naïve T cell maintenance and memory T cell differentiation. This results in a decline of naïve T cells and expansion of the pool of highly differentiated, pro-inflammatory memory T cells with an oligoclonal TcR receptor repertoire (Figure 1). In addition, naïve T cells are more prone to apoptosis with a typical age-related impairment of a pivotal intracellular phosphorylation pathway. Of note, all changes are compatible with the concept of premature ageing of the T cell system, which is at least 20 years ahead of the calendar age of individuals with normal renal function [7]. In combination with the changes in other cellular parts of the adaptive immune response (B cells and DC), this can explain the clinical observations of an impaired adaptive immune response.

Uremia leads to increased oxidative stress and inflammation, and either of them can augment the other. A leaky gut and low-grade infections like parodontitis (not shown in the figure) contribute to this process. The thymus is remarkably sensitive to both inflammation and oxidative stress, and normal age-related involution of the thymus is enhanced. In addition, the generation of lymphoid lineage precursor cells in the bone marrow can be negatively affected. As a result, the release of newly formed T cells is diminished and the number of circulating naïve T cells decreases prematurely. The reduced number of naïve T cells may reduce the T cell receptor repertoire and induce expansion of memory T cells. The memory T cell population shows expansion and differentiation to more terminally differentiated T cells with a pro-inflammatory phenotype. A previous cytomegalovirus infection may have already induced a large population of such (CD28 negative) T cells, which can further expand in the ESRD patients. The inflated memory T cell population further compromises the adaptive immune system by occupying niches (immunological space) in the bone marrow, thereby limiting the development of other T cells, a process known as limiting. All these changes weaken the adaptive immune response while promoting systemic inflammation leading to an increased risk for infections, cancer and cardiovascular events.

## 4. Oxidative Stress, Inflammation and Premature Immunological Ageing in ESRD Patients

Patients with CKD have a pro-inflammatory condition, which is reflected in an increased C-reactive protein and increased levels of pro-inflammatory cytokines and their soluble receptors [75]. The degree of inflammation is directly associated with the degree of CKD and is greatest in patients with ESRD with or without renal replacement therapy. Increased oxidative stress because of retention of uremic solutes leading to increased levels of pro-inflammatory cytokines is the most widely accepted concept [76]. However, the condition is complex as, for instance, advanced glycated proteins may activate the immune system and pro-inflammatory cytokines may generate oxidative stress. Oxidated LDL may also directly activate T cells and induce apoptosis [4]. An increase in the incidence of infectious episodes or chronic infection like parodontitis may further enhance the pro-inflammatory condition [77]. In addition, a “leaky” gut in uremia may contribute significantly to the low-grade systemic inflammation [78].

The chronic activation of the immune system leads to the combination of activated immune cells with impaired function. The latter may be a consequence of counter regulation as has been shown for instance for the response to TNFalfa, which is decreased in circulating T cells of ESRD patients [51]. Of note is that progressive uremia is associated with premature T cells ageing until ESRD is reached [65,66]. The influence of dialysis treatment, either HD or PD, once ESRD is reached, has little additional effect on T cell ageing, although PD patients may show less premature ageing [8].

Of particular interest is the observation that the thymus is remarkably sensitive to inflammation (e.g., sepsis or experimental LPS-induced thymic involution) and oxidative stress which leads to thymic involution that seems to be irreversible [79,80,81,82,83,84]. Deleterious pro-apoptotic effects on both thymus stromal cells and intrathymic T cells are underlying this phenomenon. This connection between oxidative stress/inflammation and thymic involution may explain the reduced thymic function in ESRD patients and the observation from animal experiments that loss of renal function leads to a substantial involution of lymphoid tissue like the thymus and spleen [85]. Viral infections like CMV may also lead to a modest decrease in recent thymic emigrants [60].

Again, different processes may form a vicious loop of events as loss of renal function may cause a pro-inflammatory condition, leading to thymic involution and increased susceptibility for infections, which may foster further inflammation and thymic involution. Of note, the decrease in naïve T cells may itself increase the expansion of memory T cells as has been shown in an experimental setting [86] and young adults after thymectomy as a child [87].

This places thymic function in the center of events leading up to an increased risk for typical ageing-related adverse clinical events like infections, cancer and atherosclerosis. 

## 5. Premature Ageing of the Adaptive Immune System and Clinical Events

Several studies have linked the different hallmarks of T cell ageing in ESRD patients to clinical outcomes. The expansion of highly differentiated T cells is associated with atherosclerotic disease and cardiovascular events. Highly differentiated CD4 T cells which have lost the CD28 costimulatory molecule (CD4CD28 negative or often referred to as CD4CD28 null T cells) are particularly expanded in CMV-seropositive individuals and may act as a non-traditional cardiovascular risk factor in patients with pre-existing atherosclerotic disease [88,89]. The modus operandi of these cells involves the ability to respond to fractalkine, which is a chemokine secreted by endothelial cells, invade the atherosclerotic plaque and cause plaque instability by their cytoxic effector functions (reviewed in [49]). Others reported the association between differentiated memory CD8 T cells and cardiovascular events in ESRD and individuals with unaffected renal function [8]. In concordance with these findings, reduced telomere length of peripheral blood leucocytes has been shown to be related to an increase in cardiovascular events [90,91]. The presence of highly differentiated T cells should be considered as a secondary factor promoting inflammation and cell damage in existing atherosclerotic lesions.

In kidney transplantation recipients, the expansion of highly differentiated memory CD8 T cells is associated with an increased risk of skin cancer [92] and less risk for both early acute and late rejection [93,94,95]. It can be expected that the loss of TCR repertoire in premature ageing associates with a higher risk for certain infections, but this has not been studied yet.

The other hallmark of premature ageing is the decreased thymic output of naïve T cells in ESRD patients, which is much more than can be expected from the calendar age [7]. Some studies have associated a low number of naïve T cells to an increased risk for infection, although such a relation was not consistently found after transplantation [96]. Recently, we found that a very low thymic output of naïve T cells in recipients of a kidney transplant is highly associated with all-cause mortality at follow-up [54]. Other studies have found relations between a low CD4 T cell count and mortality or infection [97,98]. As a lowered naïve CD4 T cell count is underlying CD4 T cell lymphopenia [54], these findings are in support of a major contributory role of thymic dysfunction. 

As discussed above, a loss of thymic function lowers the total number of CD4 T cells, while numbers of CD8 T cells remained relatively intact by the expansion of memory T cells. This will lead to an inversed CD4/CD8 T cell ratio, and a low CD4/CD8 T cell ratio has been recognized as a biomarker of an age-related increase in mortality [99]. This age-related low CD4/CD8 ratio is essentially different from the CMV-related lowered CD4/CD8 ratio as the latter is mainly caused by expansion of the CD8 memory T cell population [54].

The clinical data indicate that an optimal thymus function is essential for healthy ageing and underline the notion that premature immunological ageing operates on the level of the thymus and premature thymic involution is at the center of the uremia-induced changes of the T cell system. 

## 6. Interventions for Premature Immunological Ageing and Loss of Thymic Function

Recent studies have shown that thymus involution involves the ageing of the stromal microenvironment formed by thymus epithelial cells (TEC). Many factors like cytokines, sex steroids and transcription factors are likely involved in TEC ageing [29]. Expression of the TEC autonomous transcription factor FOXN1 is pivotal for differentiation and maintaining TEC integrity. A null mutation of FOXN1 in mice results in a lack of hair and thymus, and gradual excision of FOXN1 over time in an experimental model results in thymic involution [100,101]. This process can be favorably attenuated by transfecting thymic cells with FOXN1 [102].

Cellular therapy with FOXN1 producing stem cells or cytokine-to-TEC-based therapies using IL-22 or keratinocyte growth factor have shown promising results in experimental models and offer at least proof of the concept that thymic function can be (partially) restored [29].

Interleukin 7 is an important cytokine for T cell proliferation and homeostasis. Recombinant IL-7 has been used in human studies and appears to be safe. However, although peripheral T cell numbers increase, there is little direct impact on thymus function, which limits its use as a regenerative cytokine for the involuted thymus [103,104]. Of interest, targeting of IL-7 to the thymus, e.g., by a plasmid-delivered IL-7 fusion protein, was able to restore the thymus architecture and cellularity in the aged animals [105].

The transcription factor Nrf2 is a key player in the inflammatory response, as it controls a large number of pro-inflammatory genes [106]. Therefore, suppression of Nrf2 by e.g., bardoxolone methyl may ameliorate oxidative stress and inflammation in ESRD patients and favorably attenuate the effects on the immune system. 

Restoring renal function by kidney transplantation leads to a rapid clearance of inflammatory cytokines and relieves oxidative stress in ESRD patients. However, there is no reversal in any of the markers of T cell ageing even at one year after transplantation [107]. Thus, once established, thymus involution seems irreversible, leaving the ESRD patient with premature ageing at a persistent increased risk for mortality, even after regaining adequate renal function with a GFR over 60 mL/min. The underlying mechanisms are likely epigenetic changes induced by any combination of inflammation and oxidative stress associated with uremia, which are not easily reversible [1].

Of considerable interest is a recent observation that a healthy lifestyle may slow down thymus involution. Smoking and obesity are associated with fattening of the thymus [108] and an observational study showed that elderly individuals with a high intensity of daily exercise had a better preservation of thymus function and less senescence of their immune system [109,110]. Having a healthy lifestyle with sufficient exercise will likely not reverse an atrophied thymus in ESRD patients but may delay involution. Differences in lifestyle may also be part of the explanation for the substantial inter-individual variation observed at every decade of life in the number of naïve T cells and recent thymic emigrants.

## 7. Conclusions

CKD is highly associated with premature immunological ageing of the adaptive immune response, which contributes to the uremia-associated immune defect. This immune defect results in a decreased vaccination response, more infections and increased susceptibility for malignancies. The normal age-related thymus involution is accelerated because of the sensitivity of the thymus to a pro-inflammatory uremic milieu with increased oxidative stress. Maintaining thymus function may be central for an optimal functioning adaptive immune system and protection of the ESRD patient against higher morbidity and mortality. Irreversibility of thymic involution seems to be the rule, but progress has been made in the discovery of the key processes involved. This may yield in the future regenerative therapies with thymus-targeted interventions. A healthy, non-sedentary lifestyle without obesity and smoking may slow down or prevent further degeneration of thymic function and provides another reason to promote this lifestyle in every CKD patient.

## Figures and Tables

**Figure 1 toxins-12-00224-f001:**
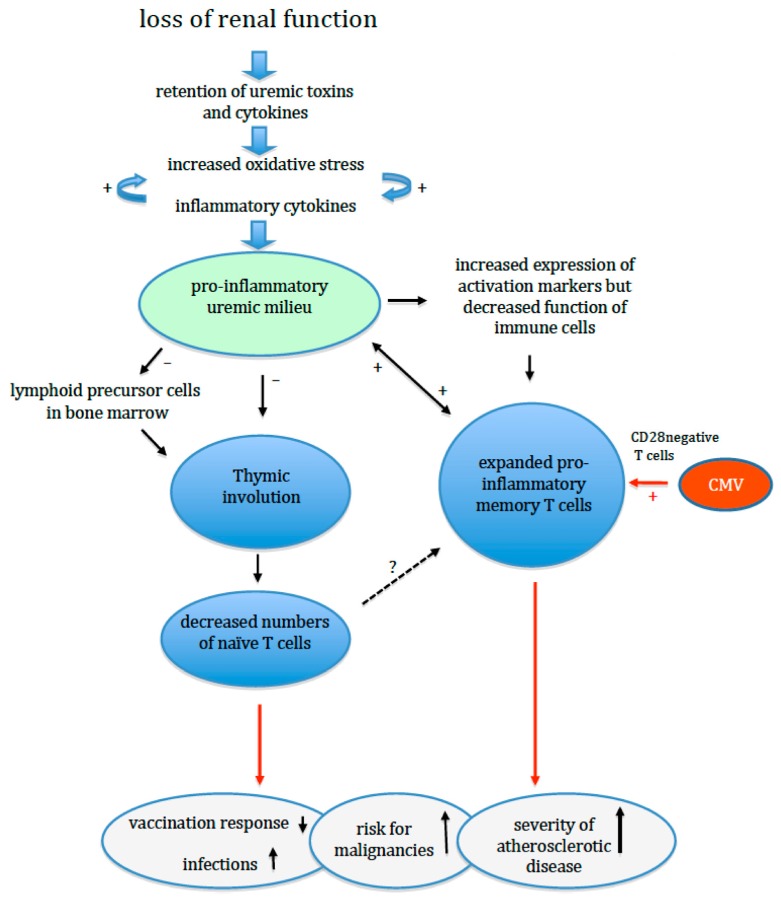
Model of uremia-associated effects on the thymus and adaptive immunity.

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
