# Peer review of "Uremia-Associated Ageing of the Thymus and Adaptive Immune Responses"

_toxins, 2020, doi:10.3390/toxins12040224_

Round 1
Reviewer 1 Report
Reviewer comments:
In this review, the authors summarize the current knowledge about the uremia-associated ageing of the thymus. They highlighted the effect of ESRD on the T cell part of the adaptive immune system. Interestingly, the authors mentioned promising results from animal experiments that have shown that rejuvenation of the thymus.
While the review is interesting and well conducted, and addresses an important phenomenon that contributes significantly to the mortality and morbidity of ESRD patients, in its current form, the manuscript requires revision before it will be suitable for publication.
Major points:
As authors mentioned, the present manuscript reviews the effect of loss of renal function on the thymus as well as naïve T cell lymphopenia and expansion of highly differentiated memory T cells, the hallmarks of immunological ageing. This is especially important since these phenomena is not reversed after restoring renal function by kidney transplantation and constitutes a long-term mortality risk factor.
1/
With respect to figure 1, the legend should say: Figure 1. “Model of” uremia-associated effects on the thymus and adaptive immunity;
since as authors mentioned; “It is not known which mechanisms contribute to the uremia-associated decrease in thymic function”.
In the same figure, the expression “increased activation and decreased function of” it's not understood.
2/
Discuss in greater depth the future prospects and potential clinical relevance of the analysis developed in the present review
Minor points:
3/
There are some typographical or grammatical errors in text:
-For example, lines 45, 123, 162, 185, 192, 197, 201, 323,
Reviewer 2 Report
In the current review paper, authors described detailed mechanism that ESRD effects adaptive immunity. The review is well-written with focusing on the topic, therefore I have only minor comments.
Comments;
- Authors discussed the associations of ESRD with adoptive immunity, but discussed those of CKD in some part (in particular in page 4). I assume that “CKD” in authors descriptions means renal insufficiency, but not nephrotic or nephritic conditions. Because the word of “CKD” broadly covers the condition in patients with renal disease, authors may need precise explanation what CKD means in some sentences.
- Introduction; page 1, “The immune system can be divided into an adaptive and innate part.” The immune system can be divided into an innate and adaptive part” sounds usual.
- Introduction; page 1, ”Collaboration between both system yields maximum effectivity of the adaptive and…” What “both system” measns?
- The immune system and ageing; page 2. “Il-6” should be changed to IL-6 as authors described IL-7 and IL-22 in the later part in page 7.
- The immune system and ageing; page 2. “Cells belonging to the adaptive immune system are B lymphocytes (B-cells) and T lymphocytes (T-cells). “ Authors already used T and B cells in page.1
- Renal failure and adaptive immunity; page 4. Authors used “CD31pos” and “CD31neg” naïve T cells, but they did as CD28negative in page 6. This should be consistent throughout the manuscript.
- Interventions for premature immunological ageing and loss of thymic function; page 8. What “near normal renal” means?
Round 2
Reviewer 1 Report
No comments
Reviewer 2 Report
Authors adequately responded for the reviewer's comments and the manuscript has been improved, therefore I have no comment in this review round.